# Towards an Enhanced, Faithful, and Adaptable Web Interaction Environment

**John Yang     Howard Chen     Karthik Narasimhan**
Department of Computer Science, Princeton University
{jy1682, howardchen, karthikn}@princeton.edu

## Abstract

We identify key areas of improvement for WebShop, an e-commerce shopping environment for training decision making language agents. Specifically, shortcomings in: 1) faithfulness of the reward function to human evaluation, 2) comprehensiveness of its content, and 3) human participation required for generating instructions has hindered WebShop's promises to be a scalable real-world environment. To solve these issues, we first incorporate greater faithfulness to human evaluation by designing a new reward function to capture lexical similarities and synonyms. Second, we identify customer reviews, similar products, and customer FAQs as missing semantic components that are most helpful to human execution of the task from surveying 75 respondents. Finally, we reformulate the attribute tagging problem as a extractive short-phrase prediction task to enhance scalability. Our `V2` reward function closes the gap between the scores of the WebShop's automated reward function (from $81.5\%$ to $87.7\%$) and human evaluation ($89.9\%$). Our attribute tagging approach achieves an accuracy of $72.2\%$ with a `t5-3b` model fine tuned on $2,000$ training data points, showing potential to automate the instruction creation pipeline.

## 1   Introduction

WebShop is a simulated e-commerce website environment for training grounded language agents on the task of purchasing a product that satisfies a given instruction [16]. Compared to previous interactive language benchmarks that are often limited by a static, non-interactive dataset or an inability to scale up [8, 14, 17], WebShop leverages large amounts of realistic data (language and other modalities like vision) and transitions scraped from the Internet to support scalable learning. A longer explanation of the WebShop environment and task can be found in §A.1.

A significant aspect of WebShop's utility towards model training is its ability to simulate real world web domains. This suggests that the WebShop environment should be realistic, scalable, and faithful to human perceptions towards this task. In this paper, we identify three key aspects where WebShop falls short on these claims, ultimately limiting its serviceability as a truly automatic environment. First, the WebShop environment does not include semantic information that heavily influences how humans perform the WebShop shopping task. Second, WebShop's original reward function consistently over-penalizes a chosen product due to its faulty exact matching criterion, compromising its faithfulness to human evaluation. Third, while WebShop's product dataset is collected in a scalable fashion via web scraping, generating corresponding instructions relies entirely on human crowd-sourcing; WebShop has $1.18$ million real products, but of these, only $12,087$ have corresponding text instructions. This reliance on human generation does not scale and bottlenecks WebShop's model training efficacy.

We put forth improvements to address these three points, demonstrating how such adjustments collectively make for a semantically richer environment that better reflects real world platforms and offer a scalable way to generate more instructions for model training. First, we solicit and incorporate

36th Conference on Neural Information Processing Systems (NeurIPS 2022).

feedback from an audience of 75 random individuals regarding information missing from WebShop that would be useful to completing the shopping task. Ensuring that WebShop captures key semantic components is fundamental to its main deliverable of constructing agents that can transfer to real-life settings. Second, we rewrite the automatic reward function's matching criteria to look for lexically similar and synonymous tokens when calculating the *attributes* and *options* score components. Our V2 reward function coheres to human evaluation much more precisely (Original $81.5\%$, V2 $87.7\%$, Human $89.9\%$). Lastly, we train and evaluate several attribute extraction models from a product's description. Our t5-3b model [12] fine-tuned on $2,000$ training points of [X=product information, Y=attributes] pairs achieves an accuracy of $72.22\%$, demonstrating the potential for high performance at an affordable cost in terms of human data collection. We then briefly discuss future plans to automate the instruction generation process. Eliminating the need for human participation in the instruction generation process is vital to WebShop's extendibility. As real world platforms evolve, WebShop's long term viability for model training hinges on how efficiently the environment, dataset, and instructions can be updated. Without such automation, WebShop's instructions and relevance will wither with time.

We believe that the collection of changes presented in this paper greatly advances WebShop's usability as an environment for designing language instructed agents with imminent real world applications, and our primary goal with this work is to make WebShop a worthwhile platform for developing web agents to the greater grounded language research community.

## 2 Related Work

Prior to WebShop, designing web-based benchmarks for grounded language agents has been studied extensively [13, 10]. This work has attempted to capture the web's scalable, semantic, interactive, dynamic, and realistic nature, but often fall short due to a relatively confined action space, an inability to scale up without human-in-the-loop feedback, or a limited set of tasks. The Mini World of Bits (MiniWoB) environment in particular has served as the test bed for a variety of approaches towards navigating and interacting with the web, such as workflow-guided exploration [7], curriculum and meta-learning [3], DOM tree representation [6], adversarial environment generation [4] and large-scale behavioral cloning [5]. However, MiniWoB's handcrafted tasks are founded on synthetic data, and its tasks do not require long-range decision making across multiple contexts. WebShop delivers on these limitations with its more diverse action and observation spaces; achieving the WebShop task requires navigating longer paths with context-based action selection and backtracking. However, WebShop under-delivers in its claims to provide a semantically rich and realistic environment, and does not deliver in its ability to scale and evolve its instructions dataset without human participation.

## 3 Environment

### 3.1 Reward Function Reformulation

WebShop's original reward function generates a composite score from calculating the similarity strictly between two products' attributes, type, options, and price, with a custom programmatic matching function per category. Exact matching is used to score attributes and options. To quantify the faithfulness of the original reward function, we randomly re-score 100 samples, selected from a pool of trajectories generated by average and expert Amazon Mechanical Turk (AMT) workers, against a human criteria. This criteria follows the original reward function with two main modifications. Instead of exact matching, points are awarded if (1) the picked product's attributes, options or type are lexically similar or synonymous with the goal's product information and (2) the desired goal value is not found verbatim anywhere in the picked product's descriptions.

The matching criteria consistently overpenalizes a picked product due to its failure to account for lexical similarities and synonyms that humans would otherwise award. For instance, given a goal token *lightweight*, the existing reward function would award neither *light␣weight* (semantically similar) nor *easy to carry* (synonym). In addition, the original approach does not reward a goal attribute or option that (1) does *not* appear in the picked product's corresponding category, but (2) does appear elsewhere in the product's description. For example, given *organic* as a desired option, a human scorer would award points if the picked product contains *organic* in its title even if *organic* is not presented as an option. The consistent disparity in the *attribute*, *options*, and *overall*

scores between the *Original* and *Human* reward functions, as shown in Figure 1, highlights the over-penalization that manifests from these discrepancies.

We implement a modified reward function that applies lexical and synonym matching for scoring attributes and options along with a comprehensive search of product information. The new proposed reward function is defined in its entirety as Equation 1. A full pseudocode description of the matching functions can be found in §A.2.1.

$$r = r_{\text{type}} \cdot \frac{match_{\text{attrs}}(U_{\text{att}}, Y_{\text{att}}) + match_{\text{opts}}(U_{\text{opt}}, Y_{\text{opt}}) + \mathbf{1}[y_{\text{price}} \le u_{\text{price}}]}{|U_{\text{att}}| + |U_{\text{opt}}| + 1} \tag{1}$$

To determine the faithfulness of the new reward function to human rewarding, we repeat the afore-mentioned verification procedure with the new reward function defined in Equation 1 and list the average scores per category in Figure 1. We also re-run imitation learning models discussed in the original WebShop paper. More details about these models can be found in §A.2.2. For both average and expert MTurk worker trajectories, the *Attribute*, *Options*, and *Overall* scores generated by the V2 reward function are all greater than the `Original` reward function scores, but do not exceed the Human benchmarks. This increase is also observed in the updated scores for IL models in Figure 2. A lengthier discussion of the advantages and shortcomings can be found in §A.2.1.

| MTurk Type | Reward | Attribute | Options | Overall |
|---|---|---|---|---|
| Average | Original | 71.7 | 50.5 | 72.4 |
| | **V2** | 74.1 | 55.0 | 74.9 |
| | Human | 75.3 | 57.0 | 76.3 |
| Expert | Original | 78.1 | 56.1 | 81.5 |
| | **V2** | 85.2 | 64.9 | 87.7 |
| | Human | 88.2 | 66.8 | 89.9 |

| Model | Reward | Score | SR |
|---|---|---|---|
| IL w/o | Original | 45.8 | 10.6 |
| LP choice | V2 | 51.7 | 11.1 |
| IL w/o | Original | 56.0 | 26.3 |
| LP Search | V2 | 60.1 | 28.1 |
| IL | Original | 59.9 | 29.1 |
| | V2 | 65.3 | 32.7 |

Figure 1: Reward function verification comparing trajectories generated by average and expert human MTurk workers.

Figure 2: Task scores and Success Rate (%) for WebShop models on original and new reward functions.

Figure 1 and 2 reflect our observation that the V2 implementation of automatic scoring reduces over-penalization and is much more faithful to human evaluation. From manual checks of 20 trajectories chosen randomly from the pool of 200 scored trajectories, the improvements in these scores can be directly attributed to the lexical and synonym matching cases. Across all 200 trajectories, there were no instances where the V2 reward function assigned a score that was greater than the corresponding Human reward function's score. The remaining gap between the V2 and Human reward functions can mainly be attributed to lexical versus numeric representations of numbers (i.e. "three" and "3") or a lack of contextualization when querying for synonyms (i.e. is "blue" used as a color or an emotion).

## 3.2 Semantic Details

We surveyed an audience of 75 individuals, each of whom were asked to (1) complete a single round of the WebShop shopping task, then (2) discuss if there was information useful for completing a shopping task that was not found in WebShop. More survey details are included in §A.3. The three most frequent responses were *customer ratings and reviews* (53 mentions), *similar products* (41 mentions), and *frequently asked questions* (37 mentions). We then implemented a *Reviews* tab on the WebShop environment that appears on a product's `item` page. Visuals are included in §A.4.

## 4 Scalability

WebShop's attribute tagging and instruction generation pipelines require human annotators. For the attribute tagging task, given a product and a pool of attributes, a human worker is tasked with assigning relevant attributes to the product. For the instruction generation task, given a product, including its title, product category, attributes, and options, a human worker is tasked with constructing a natural language query. This human-in-the-loop system is time-consuming, expensive, and also introduces potential human biases (i.e. varying degrees of knowledge across product categories). Furthermore,

this methodology lacks robustness to changes in the WebShop environment and product dataset. For instance, if new semantic signals are added to products (i.e. reviews), collecting new instructions that incorporate additional details carries a cost that must be paid every time for any future iteration. Yet, such adaptability would be crucial to WebShop's long term viability.

To automate the attribute generation task, we fine tune an out-of-box T5 model [12] to predict attributes from the product information. We train the model at different sizes on pairs of `[X=product information, Y=attributes]` drawn from WebShop's dataset of products annotated with attributes by MTurk workers. The product information consists of the title, description, and features. The corresponding label consists of a list of five attributes. We test T5 models of sizes `['small', 'base', 'large', '3b']` with training sets of size `[50, 200, 500, 1000, 2000, 3000, 4000, 5000, 6500, 8000]`. The validation and test data sets each contain $1,000$ data points. To evaluate the model's performance, we calculate accuracy as the intersection of the predictions and ground truth labels. Figure 3 plots each model's accuracy at each training set size. Additional model, training, and dataset construction details can be found in §A.5.

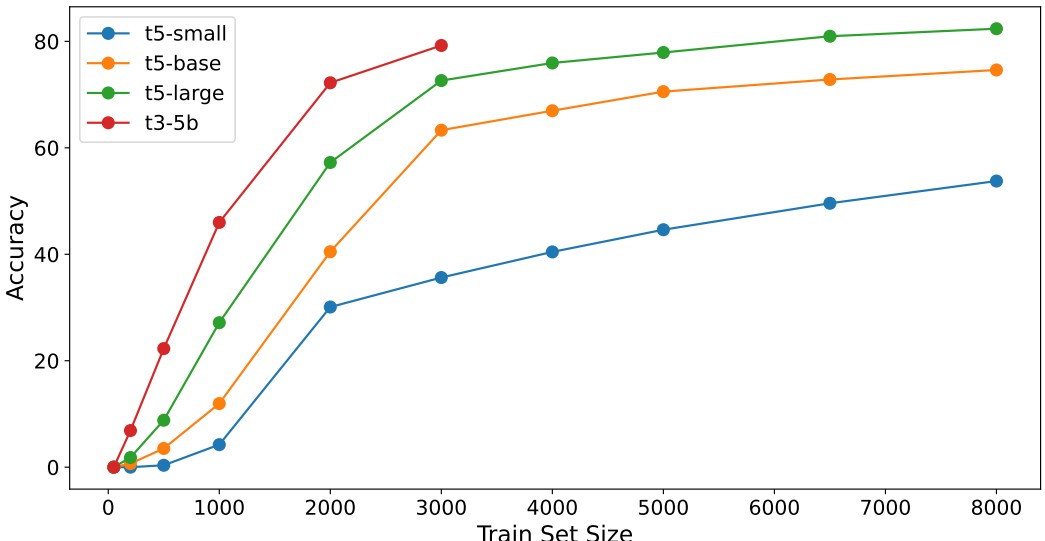

Figure 3: Performance of fine tuned T5 models of various sizes on attribution generation, reformulated as a extractive short-phrase generation task.

With $2,000$ training points, the `t5-3b` model achieves an accuracy of $72.22\%$. Larger models like `t5-large` and `t5-3b` produce structurally and syntactically sound predictions at $1000$ training points. At $2,000$ training points, `t5-3b` consistently generates a correctly structured output consisting of five unique attributes. At the same training set size, as the model size increases, accuracy increases. If this trend persists, larger models such as `t5-11b` may offer greater accuracy at an affordable cost. This reformulation demonstrates promise as an efficient and faithful replacement for human generation.

The performance of the model on attribute generation is encouraging for future work towards automating instruction generation. This model could be supplied with a product's information, attributes, options, and price, then asked to output a natural language query. However, such a model might lean towards learning more extractive practices, which in turn could confine the diversity of the outputted instructions to a finite set of learned templates. On the other hand, a text generation model with a similar set of inputs and outputs could potentially devise richer queries at the cost of requiring more human-produced training data. We briefly discuss potential task formulations in §A.6.

## 5  Conclusion

We have identified key claims where WebShop falls short, namely the environment's semantic richness, the faithfulness of the reward function to human evaluation, and the scalability of the attribute and instruction generation pipelines. Our improvements resolve key bottlenecks in WebShop's usability and makes WebShop a more fertile, solid ground for future work.

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

# A Appendix

## A.1 WebShop Background

WebShop is a simulated e-commerce website environment with $1.18$ million real world products and $12,097$ crowd-sourced text instructions. The task at hand is, given a natural language instruction describing a product, an agent or human task performer is asked to navigate a shopping site with multiple types of webpages and choose from a wide number of actions to find, customize, and purchase a product best fits the specifications outlined by the original instructions.

The WebShop environment features a variety of states (each of which corresponds to a unique web page) and a number of actions to transition from one state to another. A state *s* represents one of four types of webpages:

- `search` page: This page displays a search bar for entering search queries.
- `results` page: This page displays a list of products corresponding to a search query. Each product's product title, price, and rating are displayed.
- `item` page: This page displays a product description, which specifically includes the product title, price, rating, and buttons to view more details.
- `item_detail` page: This page shows additional information about the product depending on the page's type, which includes `description`, `features`, and `reviews`.

At each state, an agent has two choices of actions: to either *search* a text query (e.g. `search[Red shoes]`) or *choose* a text button (e.g. `choose[Size 9]`). The following Table 1 lists the full set of available actions and the state transitions they correspond to.

| Type | Argument | State $\longrightarrow$ Next State |
|---|---|---|
| search | [*Query*] | search $\longrightarrow$ results |
| choose | Back to Search | * $\longrightarrow$ search |
| choose | Prev/Next Page | results $\longrightarrow$ results |
| choose | [*Product Title*] | results $\longrightarrow$ item |
| choose | [*Option*] | item $\longrightarrow$ item |
| choose | Description/Features | item $\longrightarrow$ item_detail |
| choose | Previous | item_detail $\longrightarrow$ item |
| choose | Buy Now | Item $\longrightarrow$ Episode End |

Table 1: List of Actions in WebShop

Within the WebShop environment, an agent is then given the human-provided text **instruction** and asked to purchase a product that matches the specifications. **Rewards** are automatically computed using a combination of programmatic matching functions that consider the attributes, type, options and price of the chosen product.

Putting all these components together, the WebShop shopping task can be formulated as a partially observable Markov decision process (POMDP) $(\mathcal{S}, \mathcal{A}, \mathcal{T}, \mathcal{R}, \mathcal{U}, \mathcal{O})$ with state space $\mathcal{S}$, action space $\mathcal{A}$, deterministic transition function $\mathcal{T} : \mathcal{S} \times \mathcal{A} \to \mathcal{S}$, reward function $\mathcal{R} : \mathcal{S} \times \mathcal{A} \to [0, 1]$, instruction space $\mathcal{U}$, and a state observation space $\mathcal{O}$. Each web state $s \in \mathcal{S}$ can be rendered into a HTML observation $o_{html} \in O_{html}$ or a parallel text observation $o_{text} \in O_{text}$. The action space $A$ at each step is either an infinite space of text to search, or a finite set of buttons to click (Table 1). The transition $\mathcal{T}$ is given by the routing of the web pages as specified in Table 1.

In the WebShop environment setting, an agent is presented with a variety of challenges for language grounding, including understanding compositional instructions, query (re-)formulation, comprehending and acting on noisy text in webpages, and performing strategic exploration.

## A.2 Reward Function Details

This section of the appendix includes details regarding the implementation of the new reward function, along with several case studies of the improvements over the original reward function, along with a discussion on potential areas to refine the new function even more.

### A.2.1 Matching Implementation

The matching functions for attributes and options are implemented separately due to the selective application of lexical matching when capturing lexical similarities for options. While attributes are always lexical, options could be numeric (i.e. shoe size, dimensions, quantity/count); for options with numeric values, only exact matching is used to avoid over-scoring. For instance, if a desired option is a shoe size of 11 and the picked product specifies a chosen shoe size of 13, the lexical similarity score is high despite the option being absolutely incorrect. In the above implementation, this difference is captured by line 4 in Algorithm 2 of Figure 4; aside from line 4, the two matching algorithms are technically identical.

---

**Algorithm 1** Attribute Matching ($match_{\text{attrs}}$)

---

**Input** gAttrs, pAttrs, product
**Output** Attribute score

1: hits = 0
2: **for** g ← gAttrs **do**
3:      **if** g in pAttrs **then** hits++
4:      **for** p ← pAttrs **do**
5:         **if** fuzz(g, p)>0.85 **then** hits++
6:         **if** g in synonym(p, 5)* **then** hits++
7:      **if** g in product **then** hits++
8: **return** hits / len(gAttrs)

---

**Algorithm 2** Option Matching ($match_{\text{opts}}$)

---

**Input** gOpts, pOpts, product, optType
**Output** Attribute score

1: hits = 0
2: **for** g ← gOpts **do**
3:      **if** g in pOpts **then** hits++
4:      **if** optType is numeric **then** $break^{\dagger}$
5:      **for** p ← pOpts **do**
6:         **if** fuzz(g, p)>0.85 **then** hits++
7:         **if** g in synonym(p, 5)* **then** hits++
8:      **if** g in product **then** hits++
9: **return** hits / len(gOpts)

---

Figure 4: Implementation pseudocode for $match_{\text{attrs}}$ and $match_{\text{opts}}$. ∗ - The $synonym$ function takes two arguments: the query word and number of synonyms to return. † - If $break$ is hit, the rest of the loop (lines 5-8) is skipped and the thread of execution proceeds to the next iteration.

Figure 4 contains the pseudocode of implementations for matching attributes and options. The thefuzz [1] and PyMultiDictionary [11] modules are respectively used to determine lexical similarity and synonimity. The PyMultiDictionary library compiles a set of synonyms from educalingo.com, synonym.com, and WordNet. For each word, the most frequently occurring synonyms across all three datasets are compiled into a list, which is then statically referenced. A significant shortcoming of this library is that for words with multiple meanings, the corresponding synonyms are only of the word's most popular meaning. For instance, given the word *bat*, the synonyms returned by PyMultiDictionary all related to the act of hitting an object, rather than the animal or baseball equipment. While this is a clear downside, a small but definite advantage is that the current synonymy metric will not generate any false positives – words that PyMultiDictionary thinks are synonyms, but in reality are not. This reason accounts for why, out of all 200 trajectories scored by humans for purposes of evaluating the new reward function, no score generated by the new reward function was greater than the corresponding human reward function's score.

The following Table 2 is a list of examples, sourced from the reward verification results, where the new reward function identified and rewarded a synonym that 1. a human task worker also awarded points to during the reward verification process, but that 2. the original reward function did not award due to its exact matching criteria.

| Goal Product Value | Picked Product Value | Reason |
|:---:|:---:|:---:|
| grey | gray | Lexical Match |
| lightweight | light weight | Lexical Match |
| dry | chapped | Synonym Match |
| organic | natural | Synonym Match |
| elastic | stretchy | Synonym Match |

Table 2: Examples of matches that both human evaluators and the V2 reward function awarded, but the original reward function did not.

### A.2.2 Baseline Models

The imitation learning models that we run here are directly drawn from the baselines established in the original WebShop paper [16]. LP Search uses a pre-trained BART model to generate the search query and IL w/o LP Search uses a rule-based heuristic. LP Choice uses pre-trained BERT weights to initialize the choice action model and IL w/o LP Choice trains a Transformer from scratch.

### A.3 Task Difficulty Survey

We designed a survey with the high level goal of quantifying WebShop's usability and qualifying important gaps between WebShop's environment and real world equivalents. The survey asked the following three questions:

1. On a scale of 1 to 7, how hard or easy did you find this task? (1 - Very Hard; 7 - Very Easy)

2. In your opinion, on a scale of 1 to 7, how well did the product you chose fit the original instructions? (1 - No Relation to Instructions; 7 - Perfect Match)

3. What information would you have found helpful in completing this task that you could not find in WebShop? (Free Response)

The survey was delivered as a Google Form and distributed via a combination of posts on public online forums and a posting on WebShop's project site. Participation in the survey was completely voluntary with no compensation. Beyond the answers to the above three questions, nothing else about a user's background was collected, including the user's performance on the WebShop task, which preceded the survey question responses.

For questions one and two, users gave average scores of 5.44 and 5.75 respectively. To determine the most frequent responses for question three, we manually went through survey responses and grouped responses by common keywords. Across all 75 responses, *customer ratings and reviews* were mentioned in 53 of the replies, followed by *similar products* (41 times) and *frequently asked questions* (37 times). Beyond these, users also mentioned non-semantic enhancements and features to make task navigation easier, such as filtering/sorting products by category or price, a history of viewed products, and product recommendations. We believe pursuing implementations of these additional features would not only bolster WebShop's semantic richness and faithfulness to human task performers, but also make for a richer action space.

### A.4 Reviews Implementation

The `reviews` tab is displayed on an `item` page. Upon clicking on this tab, a list of reviews, each of which consist of a title, rating, and comment, are displayed on a separate page. From here, the user's choices of action are either to go back to the `item` page via the *back* button or go back to the search `search results` page via the *back to search* button. This is equivalent to how the `description` and `features` pages can be navigated to and from.

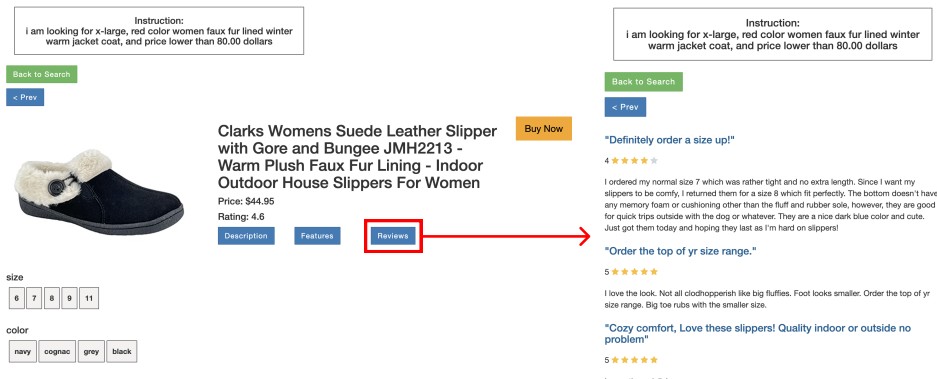

Figure 5: Reviews Page. The page can be accessed via the *Reviews* button on the `item` page

Reviews for each corresponding item were retrieved by first navigating to the corresponding product page via the product ID on `amazon.com`, then scraping the reviews section of the product page using ScraperAPI [9]. This was performed for all 1.18 million real world products in the original WebShop dataset.

## A.5 Model, Training, Dataset Construction

To construct the initial dataset for the attribute tagging model, we randomly select 10000 products from the 12087 original products with text instructions. Each of these products in this pool have attribute tags from crowd-sourcing MTurk workers, which was done by the original WebShop authors. We then reformat this data into `[product description, attribute]` pairings. We then split these values according to an $80/10/10$ split. To generate training dataset of different sizes, we sample without replacement the respective amount (i.e. `[50, 200, 500 ..., 5000, 6500]`) from the 8000 train set.

We use an out-of-box summarization model from Hugging Face's transformers library [15], following the examples laid out in the open source summarization module code to set the target model, training dataset, and validation dataset for each run.

We run the `t5-small` and `t5-base` models on a single GPU with batch size of 4, gradient accumulation step of 2, learning rate of 1e-4, and 3 training epochs. For the larger `t5-large` and `t5-3b` models, we use the `accelerate` library to handle distributed training across 4 GPUs and adjust batch size to 2, gradient accumulation step to 8, learning rate of 3e-4, and 3 training epochs [2].

We formally define the accuracy metric as follows:

$$accuracy = \frac{attributes_{predicted} \cap attributes_{truth}}{|attributes_{truth}|} \tag{2}$$

## A.6 Instruction Generation

In this section, we briefly discuss potential future work for generating natural language instructions from product information and the extracted attributes. We surmise three ways to formulate this task: populating instruction templates, summarizing a product description as an instruction, or generating product instructions from target attributes.

Creating templates is a straightforward process that would require little engineering, while guaranteeing that the output instructions are fluent and logical. A general formulation involves devising a natural language instruction template with slots that would be filled in with desired attributes, options, and a price. For instance:

> I'm looking for a <product> that is <list of attributes>. Please make sure the <option type> is <option value>, and keep it under <price>.

This simple and straightforward approach would be an immediate solution to augmenting the number of available of instructions. However, it reduces the linguistic diversity of instructions and consequently has the clear downside of drastically simplifying the challenge of query understanding.

A more promising direction that preserves linguistic diversity would be to frame creating an instruction as an purposeful extraction of product information. Similar to the formulation described in Section 4, we can define a dataset consisting of pairs of `[X=product information, Y=instruction]` taken directly from WebShop's dataset, then train a T5 based summarization model. However, due to the lack of uniformity in the crowdsourced instructions, while we expect outputs to reflect this heterogeneity, we anticipate getting such a setup to create fluent and comprehensible instructions will likely require a larger dataset that may exceed the size of WebShop's dataset of readily available, crowdsourced instructions.

Last but not least, building off of the attribute extraction model put forth in Section 4, we believe that prompting a generative model such as GPT-3 with pairs of attributes and instructions may be a viable approach that may not require a significant number of examples. However, such an approach would require calibration and fine tuning to ensure that generated instructions do not feature attributes or options that do not reflect the underlying target goal product.

