# OpenReview forum: "Towards an Enhanced, Faithful, and Adaptable Web Interaction Environment"
_NeurIPS.cc/2022/Workshop/LaReL — LaReL 2022_

### Official Review · Reviewer_KDsJ · 2022-10-18
**What about false positives?**

**Rating:** 7
**Confidence:** 3

**Review:**

The work aims to address a limitation in the scoring metric used for the existing RL benchmark environment WebShop.  They also hope to use T5 to augment the environment with additional data of automatically generated attributes.

1. I realize this is not the role of this paper but why aren't human scores in Fig 1 higher?
2. How often does the synonymy metric fail because it equates two words as the same that aren't (this is a known failing of LLMs in grounded settings -- for example, BARTScore often confuses colors and shapes)? Include some examples?
3. How much worse is (2) when analyzing the automatically generated attributes? Is this issue lessened by the use of larger LMs?

I greatly appreciate the presence of the appendix, but a little more information about the environment itself and perhaps how this change to the scoring affects models would be appreciated.

---

### Official Review · Reviewer_neCG · 2022-10-20
**Improvements to a simulated e-commerce environment**

**Rating:** 6
**Confidence:** 3

**Review:**

The authors make several improvements to the simulated website environment WebShop: a) improvement to the automatic reward function (which now includes lexical and synonym matching), b) addition of reviews to the website, c) automated extraction of product attributes via fine-tuning of T5 model (which could be used to automate the generation of instructions in the future work).

The use of surveys for the identification of features that should be added to the website (as in b)) was very much appreciated.

The following are the changes I'd like to see in the final version of the paper:
- given this work is an extension of Yai et al, 2022; there should be a section in a Background section or Appendix explaining the WebShop environment in more detail
- for section 3.1 Figure 2, explain what is meant by LP choice and LP search
- for section 3.1, add in the Appendix more examples that illustrate how and why the new reward function better matches human evaluation (and what explains the remaining difference between the reward function and human evaluation)
- for section 4, examples of instructions that can be generated using the extracted attributes, and how do those compare to human-generated queries
- for section 4 (line 135), a better explanation of the used accuracy metric

---

### Decision · Program_Chairs · 2022-10-21

Accept